Corrected: Author correction

# Adaptive individual variation in phenological responses to perceived predation levels

Robin N. Abbey-Lee [1,3] & Niels J. Dingemanse [2]

The adaptive evolution of timing of breeding (a component of phenology) in response to environmental change requires individual variation in phenotypic plasticity for selection to act upon. A major question is what processes generate this variation. Here we apply multi-year manipulations of perceived predation levels (PPL) in an avian predator-prey system, identifying phenotypic plasticity in phenology as a key component of alternative behavioral strategies with equal fitness payoffs. We show that under low-PPL, faster (versus slower) exploring birds breed late (versus early); the pattern is reversed under high-PPL, with breeding synchrony decreasing in conjunction. Timing of breeding affects reproductive success, yet behavioral types have equal fitness. The existence of alternative behavioral strategies thus explains variation in phenology and plasticity in reproductive behavior, which has implications for evolution in response to anthropogenic change.

[1] Research Group Evolutionary Ecology of Variation, Max Planck Institute for Ornithology, Eberhard-Gwinner-Str., 82319 Seewiesen, Germany. [2] Behavioural Ecology, Department of Biology, Ludwig Maximilians University of Munich, Großhaderner Str. 2, 82152 Planegg-Martinsried, Germany. [3]Present address: IFM Biology, AVIAN Behavioural Genomics and Physiology Group, Linköping University, 58183 Linköping, Sweden. Correspondence and requests for materials should be addressed to N.J.D. (email: n.dingemanse@lmu.de)

Phenotypic plasticity, the ability of organisms to adjust their phenotype to the environment, represents an important means by which organisms shift their phenology in response to environmental variation[1,2]. For example, anthropogenic change has made the climate more variable from year to year, thereby inducing selection for phenotypic plasticity in timing of breeding[2]. This adaptive evolution of population-level phenotypic plasticity requires individual variation in plasticity; therefore, predictions of population or species persistence will require insights in the ecological processes maintaining this type of variation.

One of the key factors influencing optimal timing of breeding is predation[3,4] as it is postulated as a motivator for colonial and synchronous breeding[5–8]. However, a key unresolved question is whether synchronization is an ultimate adaptive strategy (i.e., females choosing to breed at the same time as neighboring females), or rather a proximal result of a common response to an environmental cue (i.e., females all deciding to initiate breeding in the narrow window when conditions are good)[8,9]. Predators can induce seasonal increases in predation, and therefore may induce synchrony (or asynchrony) by nature of their hunting strategy. For example, the European sparrowhawk (*Accipiter nisus*) hunts by surprise attack and times its reproduction to coincide with the peak of fledging of their passerine prey, such as great tits (*Parus*

*major*)[3]. Therefore, in the presence of sparrowhawk predators, great tit females may shift their timing of breeding in order to mismatch with the peak in sparrowhawk hunting. If all females respond by starting to breed at one particular time, synchrony may increase with predator presence. In our population, great tits respond to sparrowhawk cues (calls) by decreasing time invested in conspicuous behaviors (singing), and increasing time invested in vigilance (alarm calling)[10]. Not all birds respond equally because individuals differ in heritable behaviors affecting predation risk, such as exploratory tendency assayed during short-term captivity (ranging from slow to fast behavior, with fast individuals being more likely to encounter predators and thus being more at-risk). For example, in response to predator cues, individuals improve maneuverability by decreasing body mass[11], which fast explorers (at-risk individuals) do more strongly than slow explorers[12]. Therefore, we predicted that all individuals do not respond the same to predator presence, thereby potentially decreasing breeding synchrony. We studied the consequences of these behavioral strategies for timing of breeding using replicated multi-year spatiotemporal manipulations in the wild.

We manipulated perceived predation levels (PPLs) during spring and summer (March through July) among 12 nest box plots of great tits over a 2-year period (Fig. 1a). As a low-PPL treatment, we broadcast songs of a bird species, the common

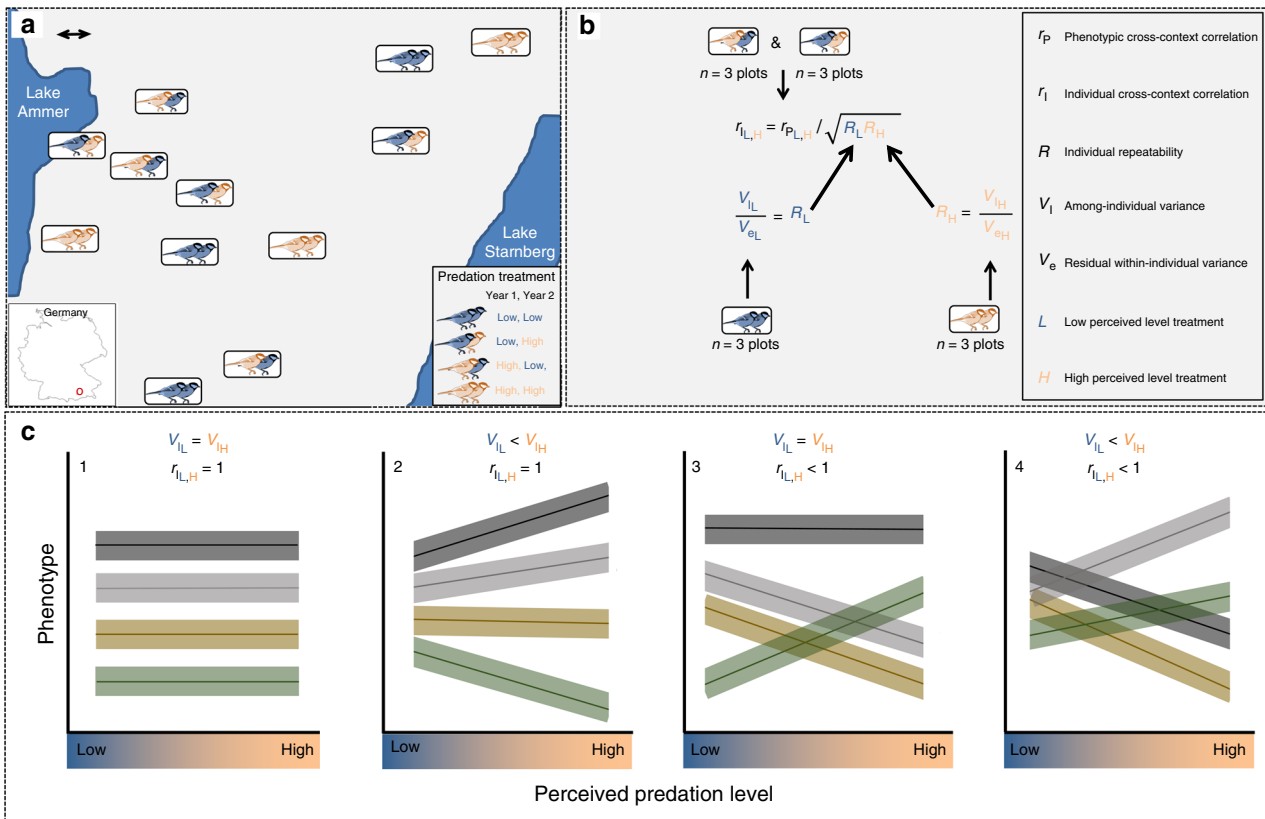

**Fig. 1** Experimental design. **a** Study area of 12 nest box plots (rectangular boxes) situated in Southern Germany. Colored great tit symbols represent each plot's treatment (blue: low perceived predation level (PPL): orange: high PPL) in the first (left-hand bird) and second (right-hand bird) year of study. Scale bar is 1 km. **b** The among-individual variance ($V_I$), the residual within-individual variance ($V_e$), and the cross-year repeatability ($R = \frac{V_I}{V_I + V_e}$) for timing of breeding (and other traits) were calculable for the low-PPL and high-PPL environment since six plots received the same treatment (3 low, 3 high) across years. The phenotypic cross-context correlation ($r_{P_{L,H}}$) between a female's timing of breeding under low versus high PPL was calculated using data from six plots that changed treatment across years. This parameter represents an attenuated estimate of the among-individual cross-context correlation ($r_{I_{L,H}}$) that our unique partial crossover study design allowed estimating (for details, see Methods). **c** This in turn enabled us to differentiate between four distinct scenarios describing how individual reaction norms for timing of breeding (and other traits) varied as a function of PPL (detailed in main text). Those scenarios differed in whether treatment-specific among-individual variance was absent ($V_{I_L} = V_{I_H}$) versus present ($V_{I_L} \neq V_{I_H}$) and whether reaction norm crossing was absent ($r_{I_{L,H}} = 1$) versus present ($r_{I_{L,H}} < 1$). Here we illustrate possible scenarios given the assumption that mean breeding date is similar in both contexts, and the variance is either the same or higher under high-PPL

blackbird (*Turdus merula*), which is neither a predator nor a competitor of great tits. As a high-PPL treatment, we broadcast calls of the European sparrowhawk. Our playback frequency matched natural vocalization frequencies, and we broadcast on a 4 day on 4-day off scheme to avoid habituation[10]. Our playback design may have also altered individuals' perception of temporal variance in risk, another factor influencing anti-predator behavior[13], although competing theories debate whether mean rather than variance is important for prey to interpret predator cues[14]. Birds decreased risky communication behaviors (detailed above) in the high-PPL treatment but actual predator numbers were not affected (based on weekly counts)[15], verifying that our manipulations influenced perceived—not actual—predation levels. In this vein, our aim with this study is to determine how behavioral types differ in their adjustment of reproductive investment in response to perceived levels.

Three plots received the low-PPL treatment in both years, three plots the high-PPL treatment in both years, and six plots changed treatment across years (Fig. 1a). This partial crossover design allowed us to estimate the statistical parameters (Fig. 1b) required for making inferences regarding how PPL treatment affected the relative timing of breeding among types of individuals (Fig. 1c). Assuming that the sparrowhawk's strategy is to produce nestlings when their prey would normally exhibit a peak in fledging production[3], PPL manipulations should increase PPL particularly for peak and late-breeding great tits. Thus, only individuals that shift to breed earlier are likely to reap a pay-off in decreased risk. Moreover, owing to optimal shifts in how vigilance-foraging trade-offs are resolved, increased PPL should also generally decrease investment in foraging, increasing relative costs of egg production, and decrease clutch size[2–4].

As a null hypothesis, we considered that birds would not adjust timing of breeding as a function of PPL (scenario 1, Fig. 1c). Alternatively, we expected that our manipulations of PPL would influence breeding decisions (scenarios 2–4, Fig. 1c). First, birds might differ in how they modify timing of breeding as a function of PPL. Sparrowhawks induce temporal variation in predation danger, therefore, individuals may shift their breeding timing to avoid the predation peak. Though responding differently, individuals may respond such that individuals breeding relatively early under low PPL would still breed relatively early under high-PPL (scenario 2). Alternatively, as at-risk individuals would benefit most from advancing timing of breeding relative to other types, only they may alter their timing of breeding, resulting in crossing reaction norms (scenarios 3 and 4) and decreased breeding synchrony with increasing PPL (as in scenario 4). We predicted that increased PPL would affect the at-risk individuals more relative to other breeders, and thus expected to see results similar to scenario 3 or 4. Previous research implies that life-history trade-offs and spatiotemporal variation in social environments equalize long-term fitness associated with slow versus fast exploration behavior in wild great tit populations[16–18]. We thus propose here that individual plasticity in timing of breeding constitutes a key adaptive component facilitating the coexistence of these alternative behavioral strategies.

We find that under low PPL, faster exploring birds breed later relative to slower exploring birds; and under high PPL the pattern is reversed and breeding synchrony decreases as a result. In addition, we find that the timing of breeding affects reproductive success, but that behavioral types have overall equal fitness by using alternative routes. We therefore conclude that these alternative behavioral strategies thus explain variation in phenology and plasticity in reproductive behavior, with implications for evolution.

## Results

**Reproductive plasticity in response to predation treatment.** Great tit females differed in how they adjusted their timing of breeding, measured by their lay date (date of clutch initiation as days since April 1) as a function of PPL treatment. The among-individual cross-context correlation between lay date expressed under low-PPL versus high-PPL ($r_{I_{L,H}} \pm SE = 0.42 \pm 0.24$, $n = 326$) was significantly below 1 (likelihood ratio test (LRT) for $r_{I_{L,H}} < 1$: $\chi^2_{0/1} = 5.53$, $P = 0.01$, Supplementary Table 1). This finding demonstrated the existence of individual differences in plasticity (see legend of Fig. 1), which came in a form where it additionally caused a reduction in breeding synchrony in the high-PPL treatment: the among-individual variance ($V_I$) in lay date was significant in both treatments (low PPL: $V_{I_L} \pm SE = 11.73 \pm 2.84$, $n = 172$, LRT: $\chi^2_{0/1} = 10.52$, $P < 0.001$; high PPL: $V_{I_H} \pm SE = 19.45 \pm 2.96$, $n = 154$, LRT: $\chi^2_{0/1} = 14.53$, $P < 0.001$) but larger in the high-PPL treatment (LRT: $\chi^2_1 = 3.86$, $P = 0.05$, Supplementary Table 2). As a consequence, cross-year repeatability ($R$; adjusted for spatio-temporal variation) was significantly reduced in the low PPL ($R_L \pm SE = 0.11 \pm 0.05$, $n = 172$) compared with the high PPL ($R_H \pm SE = 0.18 \pm 0.07$, $n = 154$) treatment (LRT: $\chi^2_1 = 4.03$, $P = 0.04$, Supplementary Table 2). By contrast, PPL treatment did not affect the population-average lay date (linear mixed-effects model (LMM): $F_{1, 22.1} = 0.00$, $P = 0.96$, Supplementary Table 3, Fig. 2a) because the effects of some females advancing were fully matched by other females delaying lay date in response to PPL (as in scenario 4, Fig. 1c).

Clutch sizes of our great tits responded differently than lay date to the treatments. Similar to lay date predictions, we predicted that individuals may alter their investment in current reproduction by changing clutch size in response to our manipulations. Either all individuals may respond the same to our manipulations, resulting in parallel reaction norms (as in scenario 1 but with negative slopes). Alternatively, predation danger may be highest only for the at-risk individuals, and therefore only these individuals would reduce clutch size. The among-individual variance in clutch size was significant in both treatments (low PPL: $V_{I_L} \pm SE = 2.04 \pm 0.31$, $n = 172$ LRT: $\chi^2_{0/1} = 25.6$, $P < 0.001$; high PPL: $V_{I_H} \pm SE = 2.07 \pm 0.36$, $n = 154$, LRT: $\chi^2_{0/1} = 13.31$, $P < 0.001$) but did not differ between treatments (LRT: $\chi^2_1 = 0.05$, $P = 0.82$, Supplementary Table 2). The among-individual cross-context correlation was tight ($r_{I_{L,H}} \pm SE = 0.84 \pm 0.12$, $n = 326$), deviating from 0: LRT for $r_{I_{L,H}} \neq 0$; $\chi^2_1 = 14.74$, $P < 0.001$, Supplementary Table 1) but not from 1 (LRT for $r_{I_{L,H}} < 1$; $\chi^2_{0/1} = 1.97$, $P = 0.07$, Supplementary Table 1), and the population-average clutch size did not differ between treatments (LMM: $F_{1, 22.4} = 0.01$, $P = 0.92$, Supplementary Table 3, Fig. 2a). Thus, females produced relatively small (or large) clutches regardless of treatment (as in scenario 1, Fig. 1c).

In other great tit populations, females breeding early also produce larger clutches[19]. The lack of congruence between individual plasticity in lay date and clutch size thus implied either that our earlier breeders did not produce larger clutches, or that PPL diminished the reproductive benefits associated with early breeding. Path analysis applied to the among-individual correlation matrix strongly supported the latter explanation (Fig. 3). The path coefficient $\beta_{\text{lay date} \rightarrow \text{clutch size}}$ was significantly more negative in the low-PPL ($\beta \pm SE$: $-0.53 \pm 0.05$, $n = 172$) compared with the high-PPL ($\beta \pm SE$: $-0.33 \pm 0.05$, $n = 154$) treatment (comparison of path coefficients between low PPL vs high PPL: $t_{325} = -2.0$, $P = 0.046$). In the high-PPL treatment, females were thus less able to reap the reproductive benefits associated with early breeding under lower PPL.

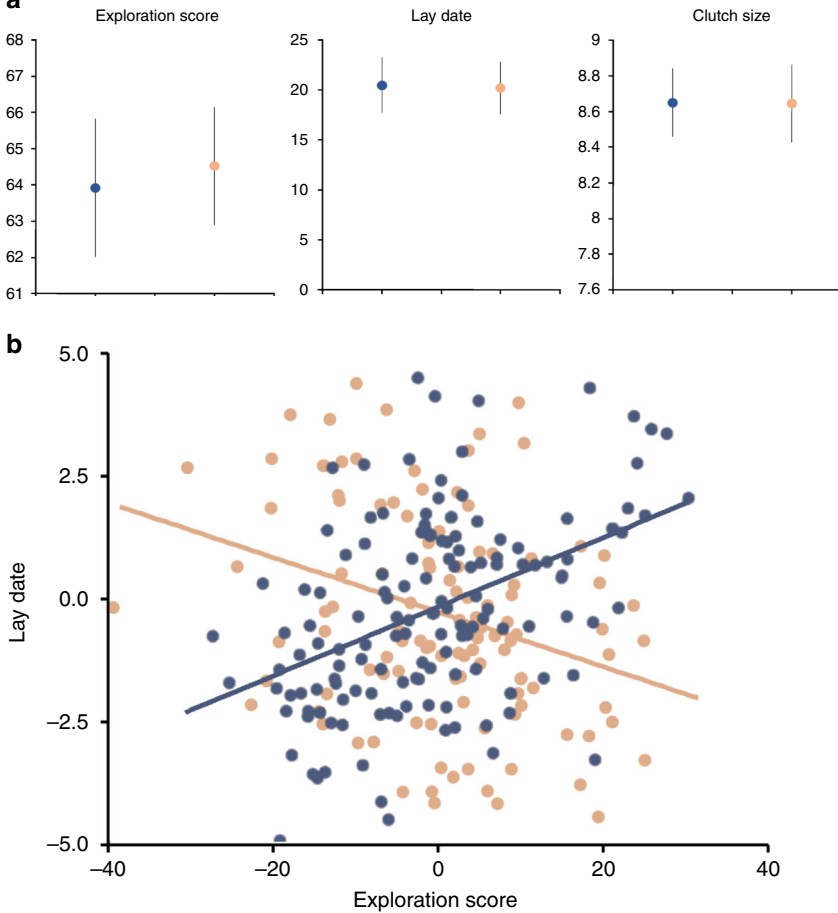

**Fig. 2** Responses to perceived predation level (PPL) manipulations. **a** Differences in traits depending on PPL exposure. Data are means with error bars representing standard error for each unique combination of treatment group and measured trait (exploration score, lay date, clutch size), illustrating the absence of an effect of treatment on mean values detected by our analyses printed in Table 1. Colored dots represent treatment (blue: low PPL: orange: high PPL). **b** Relationship between exploration score and lay date depending on PPL exposure. Points are individual's best linear unbiased predictors for lay date (y axis) and exploration score (x axis). These represent our best estimate of an individual's average value for the two focal traits corrected for the sample size per individual. Colored dots represent treatment (blue: low PPL: orange: high PPL). Source data are provided as a Source Data file, total sample size is 326 individuals, 172 in low PPL, 154 in high PPL)

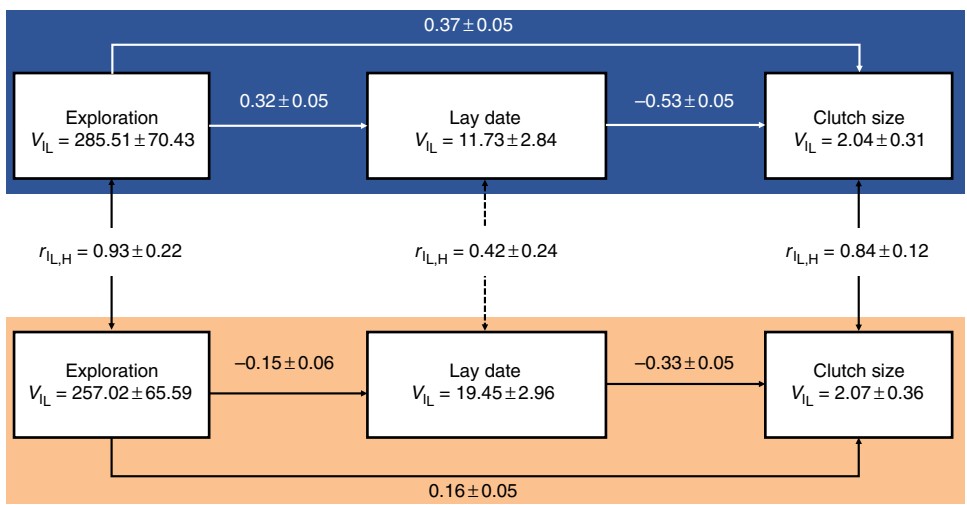

**Fig. 3** Path analyses results. Using among-individual correlation matrices to quantify the direct and indirect pathways by which exploratory behavior affected clutch size. Path coefficients (±SE) are printed alongside each hypothesized path (directional arrows) for the low-perceived predation level (PPL) (L: blue) and high-PPL (H: orange) treatment plots separately. Treatment-specific among-individual variances ($V_I$ ± SE), and cross-context correlations ($r_{I_{L,H}}$ ± SE), are printed for each trait. Source data are provided as a Source Data file, total sample size is 326 individuals (172 in low PPL, 154 in high PPL), and code is provided in Supplementary Data 1

**Behavioral rigidity in response to predation treatment**. PPL treatment affected neither exploratory tendency nor its variance components. Exploratory tendency was repeatable in both the low PPL ($R_L \pm$ SE = 0.55 ± 0.12, $n = 172$) and the high PPL ($R_H \pm$ SE = 0.52 ± 0.12, $n = 154$) treatments, significant among-individual variance occurred in both treatments (low PPL: $V_{I_L} \pm$ SE = 285.51 ± 70.43, LRT: $\chi^2_{0/1} = 8.24$, $P < 0.01$; high PPL: $V_{I_H} \pm$ SE = 257.02 ± 65.59, LRT: $\chi^2_{0/1} = 12.1$, $P < 0.001$) and did not differ between treatments (LRT: $\chi^2_1 = 0.08$, $P = 0.78$, Supplementary Table 2). The among-individual cross-context correlation between exploratory tendency expressed under low PPL versus high PPL was tight ($r_{I_{L,H}} \pm$ SE = 0.93 ± 0.22, $n = 326$), deviating from 0 (LRT for $r_{I_{L,H}} \neq 0$; $\chi^2_1 = 14.62$, $P < 0.001$) but not from 1 (LRT for $r_{I_{L,H}} < 1$; $\chi^2_{0/1} = 0.1$, $P = 0.48$, Supplementary Table 1). Population-average behavior also did not differ between treatments (LMM: $F_{1, 19.8} = 0.38$, $P = 0.54$, Supplementary Table 3). Our experiment thus showed that individuals were relatively slow versus fast explorers regardless of PPL treatment. These findings experimentally and conclusively demonstrated the existence of animal personality, defined as tight among-individual correlations in behavior across ecological contexts.

**Alternative reproductive strategies based on behavioral type**. Path analyses revealed alternative behavioral strategies, related to exploratory tendency, individual plasticity in timing of breeding in response to PPL, and consequently, the existence of repeatable individual variation in timing of breeding within each PPL treatment (Fig. 3). Under low PPL, fast explorers initiated their clutches later than slow explorers ($\beta_{\text{exploration} \rightarrow \text{lay date}} \pm$ SE: 0.32 ± 0.06, $z = 5.46$, $P < 0.001$, $n = 172$, Fig. 2b), which negatively affected their clutch size because later breeders produced smaller clutches ($\beta_{\text{lay date} \rightarrow \text{clutch size}} \pm$ SE: −0.53 ± 0.05, $z = −9.59$, $P < 0.001$, $n = 172$). However, among birds sharing the same lay date, faster explorers produced larger clutches ($\beta_{\text{exploration} \rightarrow \text{clutch size}} \pm$ SE: 0.37 ± 0.05, $z = 6.70$, $P < 0.001$, $n = 172$). Importantly, the positive direct effect of exploratory behavior on clutch size cancelled out the negative indirect effect on clutch size caused by faster explorers breeding late: the overall among-individual correlation between exploratory behavior and clutch size (Supplementary Table 4) was consequently close to 0 ($r_I \pm$ SE = 0.22 ± 0.14, $n = 172$) and not significant (LRT for $r_I \neq 0$: $\chi^2_1 = 0.23$, $P = 0.63$).

Under high PPL, fast explorers shifted forward relative to slow explorers and now initiated their clutches earlier than slow explorers ($\beta_{\text{exploration} \rightarrow \text{lay date}} \pm$ SE: −0.15 ± 0.06, $z = −2.45$, $P = 0.01$, $n = 154$) (Fig. 2b), which then positively affected their clutch size because late breeders still produced smaller clutches ($\beta_{\text{lay date} \rightarrow \text{clutch size}} \pm$ SE: −0.33 ± 0.06, $z = −5.64$, $P = < 0.001$, $n = 154$). Among individuals sharing the same lay date, fast explorers also still produced larger clutches ($\beta_{\text{exploration} \rightarrow \text{clutch size}} \pm$ SE: 0.16 ± 0.06, $z = 2.79$, $P = 0.005$, $n = 154$). Fast explorers nevertheless did not produce larger clutches overall: the among-individual correlation between exploratory behavior and clutch size (Supplementary Table 4) was again nonsignificant and close to 0 ($r_I \pm$ SE = 0.26 ± 0.19, $n = 154$, LRT for $r_I \neq 0$: $\chi^2_1 = 0.27$, $P = 0.60$).

Comparison of path coefficients revealed that PPL affected all paths by which exploratory behavior affected clutch size. First, PPL significantly advanced timing of breeding for faster relative to slower explorers ($\beta_{\text{exploration} \rightarrow \text{lay date}}$, comparison of path coefficients between low PPL vs high PPL: $t_{325} = 4.7$, $P < 0.001$). Second, PPL significantly reduced the larger number of eggs that early breeders produced ($\beta_{\text{lay date} \rightarrow \text{clutch size}}$, $t_{325} = −2.0$, $P = 0.046$). Finally, PPL significantly reduced the larger number of eggs produced by fast explorers breeding at the same date as slow explorers ($\beta_{\text{exploration} \rightarrow \text{clutch size}}$, $t_{325} = 2.1$, $P = 0.036$). Consequently, the overall among-individual correlation between exploratory tendency and clutch size did not differ between treatments (LRT comparing $r_I$ among treatments: LRT: $\chi^2_1 = 0.02$, $P = 0.89$). Importantly, we used clutch size as a proxy for reproductive fitness as it has been shown to be a reliable indicator in other populations[20,21]. An alternative measure of fitness, the number of fledglings, as expected also did not vary as a function of exploratory tendency in either treatment group (phenotypic Pearson's $r$ (95% confidence interval): low PPL: 0.04 (−0.06, 0.16), $P = 0.43$; high PPL: 0.07 (−0.04, 0.19), $P = 0.23$), implying that personality types, in fact, had equal reproductive success.

**Discussion**

Our study experimentally demonstrates individual variation in phenotypic plasticity in timing of breeding in response to PPLs, which is associated with an individualized behavioral strategy maintained by natural selection. Relatively fast-exploring birds bred relatively late when PPL was low but shifted forward to breed relatively early when PPL was high (Supplementary fig. 1). Assuming that fast explorers are at-risk individuals when predators are actually present (rather than only perceived to be present), these shifts reflect a pattern of adaptive personality-related plasticity in timing of breeding. Exploratory tendency is subject to fluctuating density-dependent selection[22]; this key mechanism is thought to explain the coexistence of avian personality types. Our finding that individual plasticity in timing of breeding represents a key component of personality-related life-history strategies thereby offers an explanation for the maintenance of individual plasticity in natural bird populations.

Early breeding increased the number of eggs produced (clutch size), but neither clutch size nor reproductive success (fledging number) varied as function of exploratory tendency despite unambiguous links between timing of breeding and personality in both treatment groups (Fig. 3). Treatment effects on how individuals resolve two interacting trade-offs can explain this apparent paradox. First, great tits face a time-allocation trade-off between foraging (resource acquisition) and avoidance of predation (vigilance)[23]. Increased investment in time allocated to predator avoidance reduces time available for resource acquisition, explaining why the reproductive benefits of breeding early diminished with increased PPL (Fig. 3). Previous work shows that slow explorers are less dominant at clumped food resources, therefore they may particularly benefit from delayed breeding to maximize resource acquisition[24]. Second, behavioral types may differ in how they resolve the trade-off between investments made in current (clutch size) versus future reproduction (longevity, onset of reproductive senescence)[25]. Fast-exploring great tits produced larger clutch sizes compared with slow-exploring great tits breeding at the same date (Fig. 3) potentially due to a faster pace-of-life[26]. However, there were no differences in the number of eggs produced between behavioral types overall, thus, in line with recent meta-analyses[27], our study does not confirm pace-of-life predictions. In line with the notion that increased investment in time allocated to predator avoidance leaves less time available to allocate towards resource acquisition, PPL also significantly reduced the larger number of eggs produced by fast explorers breeding at the same date as slow explorers. These effects combined explain why behavioral types did not differ in overall reproductive success despite exhibiting PPL-dependent differences in timing of breeding affecting reproductive success. An interesting area of future research would thus focus on directly quantifying how time and energy allocation trade-offs are resolved as a function of PPL in the wild. In addition, further

studies that also manipulate actual (rather than only perceived) predation levels are now required to discover any fitness-related consequences of reproductive behavior mis-matching the environment[28], as well as to fully address the fitness consequences associated with the alternative reproductive strategies revealed in this study. That is, while alternative personality types may indeed have equal reproductive success when predators are absent, the addition of seasonal or personality-related survival costs caused by predation may cause personality-related differences in fitness to arise when predators are present.

Specialist avian predators have previously been hypothesized to induce breeding synchrony[5,29], based on observational studies. Our study rejects this prediction experimentally, revealing that predator-induced asynchrony results from personality-related individualized responses to predation (Fig. 2b, Supplementary fig. 1). Specifically, different types may compete, with more at-risk individuals breeding earlier in the presence of more predators, allowing them to enjoy the benefits of a temporal mismatch and leaving the others to bear the brunt of the predation timed to coincide with their fledge dates. Limited previous work confirms that individuals differ in the relationship between breeding date and environmental factors, and that selection may favor more plastic individuals[30,31]. The mechanisms maintaining personality-related individual differences in phenotypic plasticity in timing of breeding will allow for some individuals to adaptively match environmental change[32], and facilitate adaptive evolution of phenotypic polymorphisms[33–35] in response to anthropogenic and other environmental change.

## Methods

**General field work procedures**. All work was ethically compliant with and carried out under Regierung von Oberbayern permit no. 55.2-1-54-2532-140-11. Data were collected in 2013 and 2014 in 12 forest plots that were established in a 10 × 15 km² area south-west of Munich, Germany[16,36,37] (Fig. 1a). Each plot consisted of 50 nest boxes arranged in a regular grid spanning approximately 9–12 ha. Lay date, clutch size, parental identities, and fledging success were monitored using standard methods (detailed in[17]). Adult exploratory behavior was measured for each captured parent when nestlings were 7 or 9 days old, using a cage test adapted from a classic novel environment test[16,38,39]. See[36] for a full description of the procedure. Briefly, each individual was recorded for 2 min; the sum of movements between different locations (scores ranged from 2 to 130) was used as a proxy of exploratory behavior. Values were scored later from the recording by an observer blind to the subject's identity and treatment. This exploration score is a measure of activity that correlates with anti-predator boldness in our population[36; thus we have validated its use as a proxy for risk-taking behavior in the face of predation threat. We performed 607 tests on 497 unique (ringed) birds. Of these, 387 were tested in only 1 year and 110 were tested in both years. Of the 110 birds with repeat measures, 29 individuals received the predator treatment both years, 32 received control both years, and 49 received both treatments.

**PPLs experiment**. We conducted a playback experiment in order to manipulate PPLs (see[10] for full details). Four speakers (Shockwave, Foxpro, Pennsylvania, USA) were evenly distributed across each plot in February and removed in July. For the first year of treatment (2013), assignment of treatments to plots was randomized, with the constraint that there be no initial differences between treatments in average breeding density, lay date, latitude, or longitude based on data from previous years. Six plots received a low-PPL treatment and six plots received a high-PPL treatment in the first year; the treatment was switched in half of the plots for the second year (Fig. 1a). Assignment of treatments to plots was again randomized, conditional on the same constraints detailed above. In low-PPL plots, speakers were programmed to play songs of a sympatric, non-predator avian species, the Eurasian blackbird (*Turdus merula*). In high-PPL plots, speakers were programmed with calls from sparrowhawks (*Accipter nisus*; a sympatric, avian predator species). Bird sounds were acquired from the Xeno-Canto database (www. xeno-canto.org) or provided by H. H. Bergmann. All speakers were programed to match the normal vocalization of our playback species: speakers broadcast approximately 60% of the time during the first 3 h after dawn and the last 3 h before dusk (six 6-min song/call bouts per hour) and speakers broadcast approximately 15% of the time during the rest of the daylight hours (1.5 bouts per hour). The amount of silence between playback bouts was determined randomly to avoid habituation. Playback was broadcast at 90 dB (intensity was set to match the normal intensity of bird songs and calls and was measured at 1 m with a sound level meter). Sparrowhawks are resident predators—they stay in the area of their

nest and hunt over a wide territory surrounding it[40]. This means that presence of a sparrowhawk during the breeding season (like our sound cues) should signal potential predation risk throughout the rest of the season to prey. In addition, a single observation of a predator is known to have a long lasting effect on prey, as it is more difficult to assess absence of a predator, whereas the costs of mis-assessment are higher[14]. Therefore, playback was given for 4 consecutive days (on), followed by 4 consecutive days of non-playback (speakers were off), the cycle was repeated throughout the season; this design prevented habituation[10] without decreasing the effectiveness of the high-PPL manipulation. Data were not biased by dispersal events as only 0.13% of birds (two individuals) have moved between plots in our years of collecting data (2010–2017), and no birds moved between treatment plots during the years of our study.

**Comparing trait means across treatment groups**. We used univariate mixed-effects models to determine whether PPL treatment affected the population-mean trait value, fitting either lay date (defined in days from 1st April), clutch size, or exploratory behavior, as the focal response variable (Supplementary Table 3). Here, treatment was fitted as a two-level categorical variable (low vs. high PPL). Random intercepts were further fitted for the unique combination of plot and year (Plot-Year; $n = 12$ plots × 2 years = 24 levels) as treatment varied at this level, thereby avoiding pseudo-replicated values of $P$ for effects of treatment[10,15,18]. Random intercepts were also fitted for individual identity, where female (rather than male) identity was assigned in analyses of lay date and clutch size because our previous work showed that female rather than male identity determines such life-history traits[18]. Exploratory behavior, by contrast, was measured for each individual parent separately, and individual identity effects thus estimated using data from both sexes. In some populations, exploratory behavior varies between sexes and over the course of the day[39]; the analysis of this behavior therefore also included sex (fitted as a two-level categorical variable: female vs. male) and time of day (hours from sunrise; fitted as a continuous variable; mean centered and expressed in standard deviation units) as two additional fixed effects. These univariate analyses thus partitioned the total phenotypic variance ($V_P$; subscript P for phenotypic) not attributable to fixed effects into variance among PlotYears ($V_S$; subscript S for spatiotemporal), variance among individuals ($V_I$; I for individual), and residual within-individual variance ($V_e$; e for error):

$$V_P = V_S + V_I + V_e \qquad (1)$$

Values of adjusted repeatability ($R$) were calculated for each random effect as the variance attributable to the focal effect divided by the total phenotypic variance not attributable to fixed effects ($V_P$)[41].

The significance of fixed effects was based on the $F$-statistic and numerator and denumerator degrees of freedom from the algebraic algorithm in ASReml 3.0[42]. Statistical significance of a random effect was calculated using a LRT where this $\chi^2$-distributed test statistic was estimated as twice the difference in log likelihood between the full model and a model with the focal random effect removed[43–45]. For variances (random effects), the value of $P$s was calculated assuming an equal mixture of $\chi^2(0)$ and $\chi^2(1)$ because variances are bound to be zero or positive[46–48] (denoted by $\chi^2_{0/1}$ in our statistical tables).

**Patterns of individual variation in reaction norms**. We used bivariate mixed-effects models to estimate the pattern of among-individual variation in plasticity in response to PPL treatment for each of the three phenotypic traits (lay date, clutch size, exploratory behavior; defined above) separately (Supplementary Tables 1, 2). Each bivariate mixed-effects model fitted the focal trait expressed in the low-PPL versus high-PPL treatment as two separate response variables (e.g., lay date expressed in the low-PPL treatment, and lay date expressed in the high-PPL treatment). The intercept values of the two response variables represented the treatment-specific mean values, and no further fixed effects were thus included (except for analyses of exploratory behavior fitting sex and time of day as fixed effects, see above). Random intercepts were included for PlotYear and individual identity (as above). These bivariate analyses thus partitioned the total phenotypic variance not attributable to fixed effects ($V_P$) into variance among PlotYears ($V_S$), variance among individuals ($V_I$), and residual within-individual variance ($V_e$) similar to the univariate models detailed above (Eq. 1) but each variance component was now estimated for the low PPL (L) and high PPL (H) separately:

$$V_{P_L} = V_{S_L} + V_{I_L} + V_{e_L} \qquad (2)$$

$$V_{P_H} = V_{S_H} + V_{I_H} + V_{e_H} \qquad (3)$$

This formulation of the data enabled us to test whether a focal variance component differed between treatment groups (Supplementary Table 2). The statistical significance of treatment effects on a focal variance component was estimated using a LRT, calculated as twice the difference in log likelihood between the full model (estimating treatment-specific variance components), and a model where the focal random effect of interest was constrained to be identical across treatment groups[49]. The associated value of $P$ was calculated assuming 1 degree of freedom ($\chi^2_1$ in Supplementary Table 2). We used this approach to test whether the among-individual variance was the same ($V_{I_L} = V_{I_H}$; scenarios 1 and 3; Fig. 1) or different ($V_{I_L} \neq V_{I_H}$; scenarios 2 and 4; Fig. 1) among treatment groups. We also used this approach to test whether individual repeatability ($R_I$) differed among

treatment groups, which was achieved by running the same test on variance standardized data[49] (Supplementary Table 2).

These bivariate mixed-effects assumed a bivariate normal distribution estimating all level-specific variances ($V$) and covariances (Cov):

$$\Omega_S = \begin{bmatrix} V_{S_L} & \mathrm{Cov}_{S_{L,H}} \\ \mathrm{Cov}_{S_{L,H}} & V_{S_H} \end{bmatrix} \qquad (4)$$

$$\Omega_I = \begin{bmatrix} V_{I_L} & \mathrm{Cov}_{I_{L,H}} \\ \mathrm{Cov}_{I_{L,H}} & V_{I_H} \end{bmatrix} \qquad (5)$$

$$\Omega_e = \begin{bmatrix} V_{e_L} & \mathrm{Cov}_{e_{L,H}} \\ \mathrm{Cov}_{e_{L,H}} & V_{e_H} \end{bmatrix} \qquad (6)$$

Importantly, treatment varied among plots within years but not within plots within years. The covariance between the low-PPL and high-PPL treatments could therefore not be estimated at the PlotYear level and was consequently constrained to zero ($\mathrm{Cov}_{S_{L,H}} = 0$)[49]. Similarly, within each year, each individual experienced a single treatment; the within-individual covariance between the same trait expressed in the low-PPL and the high-PPL treatments was thus also not open to estimation and consequently constrained to 0 ($\mathrm{Cov}_{e_{L,H}} = 0$)[49]. Owing to plots switching treatments across years (Fig. 1a), we acquired phenotypic data of the same individuals subjected to both low-PPL and high-PPL treatments, implying that the among-individual cross-context covariance ($\mathrm{Cov}_{I_{L,H}}$) of interest was open to estimation (Fig. 1b).

Covariances are presented as standardized correlation coefficients ($r$) calculated as $r_{x_{L,H}} = \mathrm{Cov}_{x_{L,H}} / \left( \sqrt{V_{x_L} V_{x_H}} \right)$, where $x$ represents the focal hierarchical level of interest. The phenotypic correlation in the data between measurements of a focal trait in the low-PPL versus the high-PPL treatment ($r_{P_{L,H}}$) was calculated as:

$$r_{P_{L,H}} = r_{S_{L,H}} \sqrt{\frac{V_{S_L}}{V_{P_L}} \frac{V_{S_H}}{V_{P_H}}} + r_{I_{L,H}} \sqrt{\frac{V_{I_L}}{V_{P_L}} \frac{V_{I_H}}{V_{P_H}}} + r_{e_{L,H}} \sqrt{\frac{V_{e_L}}{V_{P_L}} \frac{V_{e_H}}{V_{P_H}}} \qquad (7)$$

where $r_{S_{L,H}}$ represents the among-PlotYear cross-context correlation, $r_{I_{L,H}}$ the among-individual cross-context correlation, and $r_{e_{L,H}}$ the within-individual cross-context correlation. As $\mathrm{Cov}_{S_{L,H}} = 0$ and $\mathrm{Cov}_{e_{L,H}} = 0$ were both 0 (see above), $r_{S_{L,H}} = 0$ and $r_{e_{L,H}} = 0$ were also both 0, implying that Eq. (7) can be simplified into:

$$r_{P_{L,H}} = r_{I_{L,H}} \sqrt{R_{I_L} R_{I_H}} \qquad (8)$$

where $R_{I_L}$ and $R_{I_H}$ represent the adjusted individual repeatabilities for each treatment group estimated as $R_{I_L} = V_{I_L}/V_{P_L}$ and $R_{I_H} = V_{I_H}/V_{P_H}$, respectively. Calculation of the among-individual cross-context correlation ($r_{I_{L,H}}$) consequently only required information of the phenotypic cross-context correlation ($r_{P_{L,H}}$) and treatment-specific repeatabilities:

$$r_{I_{L,H}} = r_{P_{L,H}} / \sqrt{R_{I_L} R_{I_H}} \qquad (9)$$

This equation (presented in Fig. 1b) demonstrates that the phenotypic correlation between two labile traits represents an attenuated estimate of the among-individual correlation when the within-individual correlation is 0 by design[49,50]. This key parameter was estimable because treatments were allocated using our unique partial crossover design, enabling estimation of all underlying components (Figs. 1a, b).

The statistical significance of the among-individual cross-context correlation was assessed using a LRT, calculated as twice the difference in log likelihood between the full model and a model where $\mathrm{Cov}_{I_{L,H}}$ was constrained to the value 0 (Supplementary Table 1). The associated value of $P$ was calculated assuming 1 degree of freedom ($\chi^2_1$ in Supplementary Table 1). Differentiating between the four distinct scenarios (presented in Fig. 1c) required testing whether $r_{I_{L,H}}$ deviated from the value one. This was achieved by using an LRT, calculated as twice the difference in log likelihood between the full model and a model where $r_{I_{L,H}}$ was constrained to the value 1. The value of $P$ was calculated assuming an equal mixture of $\chi^2(0)$ and $\chi^2(1)$ because correlations deviating from the value 1 can do so only by being lower (not higher) than 1[46–48] ($\chi^2_{0/1}$ in Supplementary Table 1).

**Path analyses**. We used a tri-variate version of the mixed-effects model detailed above to estimate among-individual correlations ($r_I$) between lay date, clutch size, and exploratory behavior, and performed this separately for the low-PPL and the high-PPL treatments. Those tri-variate models included the same fixed and random effects structures as detailed for the bivariate models; as both models estimated covariances among three traits expressed within the same environment, among-individual correlations were calculated from a model estimating all level-specific covariances (Supplementary Table 4). Path analysis was subsequently applied to the among-individual correlation matrix estimated for each treatment group separately (Supplementary Table 4). The sem package in R was used to

calculate path coefficients (plus standard errors) associated with a model simultaneously hypothesizing that exploratory behavior affected clutch size directly, as well as indirectly by affecting lay date (Fig. 2). The value of $P$ associated with each path was calculated using $z$-tests. We used $t$-tests to compare estimates from the two treatment groups.

**Reporting summary**. Further information on experimental design is available in the Nature Research Reporting Summary linked to this article.

## Data availability
All data generated or analyzed are included in this published article and its supplementary information files. The source data underlying all results, figures, and supplementary figures are provided as a Source Data file.

## Code availability
Example code for all analyses are included as Supplementary Data 1.

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

## Acknowledgements
All work was ethically compliant with and carried out under Regierung von Oberbayern permit no. 55.2-1-54-2532-140-11. We thank H.H. Bergmann for providing the blackbird songs, and R. Bijlsma, J. van Diermen, W. Forstmeier, and H. Knuewer for input on the experimental design. We are grateful to J. Wijmenga and K.J. Mathot (planning and preparing for the experiment), A. Mouchet and M. Moiron (field work coordination), P. Sprau (preparation of recordings), J. Brommer, F. Santostefano and P. Niemelä (statistical analyses), and members and students of the research group Evolutionary Ecology of Variation (field data collection). We thank J. Brommer, S. Patrick, D. Westneat, and J. Wright for feedback on the manuscript. N.J. Dingemanse was funded by the Max Planck Society and by the German Science Foundation (grant no. DI 1694/1-1)

## Author contributions
R.N.A.-L. and N.J.D. conceived the study idea and experimental design, analyzed the data, and wrote the manuscript together.

## Additional information

**Competing interests:** The authors declare no competing interests.

