## [Peer Review File · Nature Communications]

Reviewers' Comments:

Reviewer #1:

Remarks to the Author:

The paper by Abbey-Lee and Dingemanse reports the results of a study designed to explore the relationships among individual differences in exploratory behavior, plasticity in the timing of reproduction in response to perceived predation risk and fitness in a wild population of great tits. This paper is related to the growing literature concerned with the adaptive significance of individual differences in behavior in the wild. Great tits have been a model organism for studying this question, and there is well-characterized repeatable and heritable variation among individual great tits in exploratory behavior, i.e. how individual birds differ in the way they explore their environment.

There is currently a lot of interest in the ultimate factors that can maintain behavioral variation – why aren't all birds slow explorers, for example? This paper provides an intriguing possible answer to this question, which is that behavioral types differ in how their reproductive strategy is influenced by the environment (perceived predation risk), and different behavioral types of individuals alter their timing of reproduction in a systematic way – slow explorers breed early when it's safe, and breed late in the season when it's risky, while fast explorers do the opposite. However, neither strategy is better than the other, because at the end of the day both types of birds end up with similar clutch sizes. This suggests that the different behavioral strategies have equal fitness in the wild, which is why the variation can persist.

There is a desperate need for long term field studies that document the fitness consequences of behavioral variation because an outstanding question in the animal personality literature has to do with whether individual behavioral variation persists over time and, if so, what are the selective factors that keep one behavioral type from becoming fixed in the population. Therefore this study has the potential to be a very important contribution to this burgeoning literature. One of the things that differs between this paper and most of the literature to date is that it combines both population monitoring over multiple years in the field with experimental manipulation of the environment (in this case, perceived predation risk by sparrow hawks). This merging of both observational and experimental approaches has the potential to be very powerful but of course it also has its own share of drawbacks, namely that it immediately raises questions about the ecological relevance of the experimental manipulation (a drawback of the experimental approach), and with the lack of experimental control over the observational study.

For example, while the authors went out of their way to prevent habituation to the sparrowhawk calls, in doing so they might have created differences in the perception of certainty rather than differences in the perception of risk between the high vs low predation plots. Birds in the control group were probably more certain that it was safe while birds in the experimental group were given contradictory information – sometimes it was really dangerous and for the next four days it was safe. Therefore the differences between the two treatments might more reflect perceptions of certainty rather than perceptions of risk per se. This could influence the authors' interpretation of the behavioral strategies.

Similarly, an issue with the lack of experimental control over the field study is that the authors presumably had no means to prevent birds from moving between their carefully-defined plots. I might have missed it but I couldn't find anywhere in the paper where the authors tell the reader whether birds moved between plots. Did birds leave the risky plots? If so, was movement related to behavioral type?

There are a number of other issues with the study and with the paper that the authors should consider.

First and foremost, the authors should show the data rather than just the stats. They show figures for the predicted results (Figure 1) and for their interpretation of the results (Figure 2) but do not actually show the data, instead relying on the reader to infer the patterns from the stats in Table 1. Without seeing the raw data themselves it is difficult to get a sense of the distribution of the data and the strength of the patterns.

Second, the authors rely on clutch size as a proxy for fitness – this needs to be justified. Is a larger clutch size always great fitness, and might that depend on perceived predation risk? What if females prioritize quality of eggs over quantity of eggs?

Here are more minor comments:

Line 28: plasticity of what?

Line 49: “starting to breed”

Line 51: choose another word besides “down-regulating” – something like decreasing (same for “up-regulating”)

Line 60: omit “of”

Line 64: I know it’s in the methods but tell the reader here how often and for how long you broadcast the songs. Although it looks as though the authors took measures to prevent habituation of the great tits to the songs of the sparrow hawks, did the treatment regime reflect the biology of sparrow hawks? For example calls were played for 4 consecutive days then not played for the next 4 consecutive days. Is this how sparrowhawks would normally behave? What information is this experimental regime providing to the great tits about danger? Is the treatment manipulating risk or is it more influencing uncertainty about risk? The authors clearly assume the former – that they are manipulating perceived risk – but it would be worthwhile to consider the alternative, i.e. that the “high risk” treatment was really the “uncertain” treatment.

Line 75: “game”? Do you mean “strategy”?

Line 115: remove “,” after “Either”

Reviewer #2:

Remarks to the Author:

This novel and impressive study employed a large-scale manipulation to test how environmental variation drives plasticity in reproductive strategies, and whether plasticity differs by personality type. The authors found that in great tits, perceived risk of predation influenced variation in breeding phenology, and that this variation was explained by personality type (exploratory speed). The experimental design is elegant, and the sample sizes were sufficiently large to enable estimation of within and among individual variation. The results provide important new insights into the evolution of personality, and how the risk of predation generates variation in plasticity.

The one major suggestion that I have is to more clearly differentiate between the various aspects of reproductive investment and breeding decisions that were measured, and their potential fitness

outcomes. In the discussion, clutch size, which is framed as a measure of reproductive investment (e.g., line 114) is conflated with reproductive success (also defined as clutch size on line 192). Conceptually, this is problematic because the premise of this study rests on the idea that the offspring of late breeders will face a greater risk of predation. Because of this, clutch size will not be a good measure of reproductive success, since the fitness benefit of, for example, a 5-egg clutch laid early in the season will on average be higher than that of a 5-egg clutch laid later in the season, at least in the presence of predators.

Because this experiment separated perceived risk of predation from actual predation risk, the actual fitness cost of breeding in a high-predation environment will not have been realized by individuals in the high-PPL group. Thus, their reproductive success may track more closely with their clutch size than individuals in a truly high-predation environment. But regardless, distinguishing between reproductive investment and reproductive success in interpreting these results is important for understanding how fluctuating selection on plastic responses to environmental conditions may maintain distinct personalities. It seems that fledging success was measured in this study (lines 227-228); it would also be nice to see those relationships discussed here.

Additionally, while I realize the manuscript has tight length constraints, I think a bit more discussion in two areas would help to place these findings in context more clearly. First, faster-exploring birds are described as having a faster pace-of-life, based on previous work. But the results shown here don't seem to be entirely consistent with this. Faster explorers lay more eggs than slow explorers breeding on the same date, but if I'm interpreting these results correctly, they don't lay more eggs overall – suggesting that their annual reproductive investment is not higher on average. Perhaps they do invest more at later stages of breeding (and achieve higher average annual reproductive success, offsetting their shorter lifespans), but this isn't clear from the discussion presented here.

Second, I think a bit more discussion on why slow explorers might delay their average lay date (as opposed to just their lay date relative to fast explorers) under high-PPL would improve the clarity. Competition is briefly mentioned in general terms on line 214, but is there evidence that resource competition between slow and fast explorers is a likely driver of the delay in lay date amongst slow explorers in high-PPL?

A few other minor suggestions:

- The scenarios in Figure 1c all keep the mean breeding date similar under both contexts, and the variance either the same or higher under high-PPL. Of course it isn't possible to graphically illustrate all possible scenarios, but I would recommend stating that this figure illustrates possible scenarios given those assumptions. The analyses done should enable the authors to distinguish among these and other scenarios (e.g., reduced variance in lay date under high PPL or an overall advancement in lay date), so this is not at all a criticism of the analytical framework, simply a suggestion to clarify the possible patterns.

- The term "April date" is used on line 98 – something like "days since April 1" would be clearer

- The rates at which predator vocalizations were broadcast seems high – do sparrowhawks really vocalize that much (lines 253-256)?

- Line 293: It looks like "was" should be deleted from the sentence "...between the full model and a model with the focal random effect removed"

- Line 392: "is" should be "are"

- Line 193: "did not vary as a function"

Reviewers' comments:

Reviewer #1 (Remarks to the Author):

The paper by Abbey-Lee and Dingemanse reports the results of a study designed to explore the relationships among individual differences in exploratory behavior, plasticity in the timing of reproduction in response to perceived predation risk and fitness in a wild population of great tits. This paper is related to the growing literature concerned with the adaptive significance of individual differences in behavior in the wild. Great tits have been a model organism for studying this question, and there is well-characterized repeatable and heritable variation among individual great tits in exploratory behavior, i.e. how individual birds differ in the way they explore their environment.

There is currently a lot of interest in the ultimate factors that can maintain behavioral variation – why aren't all birds slow explorers, for example? This paper provides an intriguing possible answer to this question, which is that behavioral types differ in how their reproductive strategy is influenced by the environment (perceived predation risk), and different behavioral types of individuals alter their timing of reproduction in a systematic way – slow explorers breed early when it's safe, and breed late in the season when it's risky, while fast explorers do the opposite. However, neither strategy is better than the other, because at the end of the day both types of birds end up with similar clutch sizes. This suggests that the different behavioral strategies have equal fitness in the wild, which is why the variation can persist.

There is a desperate need for long term field studies that document the fitness consequences of behavioral variation because an outstanding question in the animal personality literature has to do with whether individual behavioral variation persists over time and, if so, what are the selective factors that keep one behavioral type from becoming fixed in the population. Therefore this study has the potential to be a very important contribution to this burgeoning literature. One of the things that differs between this paper and most of the literature to date is that it combines both population monitoring over multiple years in the field with experimental manipulation of the environment (in this case, perceived predation risk by sparrow hawks). This merging of both observational and experimental approaches has the potential to be very powerful but of course it also has its own share of drawbacks, namely that it immediately raises questions about the ecological relevance of the experimental manipulation (a drawback of the experimental approach), and with the lack of experimental control over the observational study.

Response: We are very happy with the positive judgement of the quality of our work, and the merit of performing experimental field research to address key outstanding questions in personality research. Thank you very much for your time thoughtfully reviewing our manuscript.

For example, while the authors went out of their way to prevent habituation to the sparrowhawk calls, in doing so they might have created differences in the perception of certainty rather than differences in the perception of risk between the high vs low predation plots. Birds in the control group were probably more certain that it was safe while birds in the experimental group were given contradictory information – sometimes it was really dangerous and for the next four days it was safe. Therefore the differences between the two treatments might more reflect perceptions of

certainty rather than perceptions of risk per se. This could influence the authors' interpretation of the behavioral strategies.

Response: The reviewer brings up an interesting concern regarding the nature of our manipulation. We are certain that the birds should have perceived our treatment as a manipulation of mean rather than variance in predation levels. This is because in our system, prey do not typically see predators on a regular or daily cycle. Sparrowhawks are resident predators – they stay in the area of their nest and hunt over a wide territory surrounding it. This means that presence of a sparrowhawk during the breeding season (like our sound cues) should signal potential predation risk throughout the rest of the season to prey. Additionally, sparrowhawks don't vocalize while hunting, so the current absence of cues does not mean that the area is currently safe. Finally, previous research implies that a single observation of a predator has a long-lasting effect. Animals don't seem to really use the variance, but rather assume presence for a long time if there is any signal of a predator (see e.g. Gabriel et al. 2005, *Am Nat* 166:339-353). In response to this comment, we implemented textual clarification in lines 288-295.

Similarly, an issue with the lack of experimental control over the field study is that the authors presumably had no means to prevent birds from moving between their carefully-defined plots. I might have missed it but I couldn't find anywhere in the paper where the authors tell the reader whether birds moved between plots. Did birds leave the risky plots? If so, was movement related to behavioral type?

Response: Great tit breeders are very site faithful (e.g. Harvey et al. 1979, *J Anim Ecol* 48: 305-313), and birds in our population are no exception. Looking at our breeding season data over our long-term monitoring period (2010-present, data was not available yet for last year, 2018, when we ran these calculations) we only found two instances of individuals moving between our plots. One of these occurred before our experiment. Only one bird moved during our experiment, and she moved from one predator treated plot to another predator treated plot. These 2 birds were out of a total of 1454 observations of repeat breeding birds, i.e. 0.13%. We agree that we expected that birds may potentially leave risky plots, but we found no evidence of this in our breeding season treatment, nor in a separate experiment performed during winter (Abbey-Lee et al. 2016, *Behav Ecol* 27:857-864). See text additions lines 295-297.

There are a number of other issues with the study and with the paper that the authors should consider.

First and foremost, the authors should show the data rather than just the stats. They show figures for the predicted results (Figure 1) and for their interpretation of the results (Figure 2) but do not actually show the data, instead relying on the reader to infer the patterns from the stats in Table 1. Without seeing the raw data themselves it is difficult to get a sense of the distribution of the data and the strength of the patterns.

Response: Thank you for this comment. We fully agree that it is important to strengthen our conclusions by supplementing our statistical results with fitting illustrations based on

raw data. We have thus created two new figures. Both are currently incorporated into a new figure 2. Our previous Figure 2 is now Figure 3. And our previous Figure 3 has been moved to supplemental materials to keep the number of total figures in the manuscript the same.

In Figure 2a, we now print the mean (plus standard error) for each unique combination of treatment group and life-history trait (lay date, clutch size, fledge success), thereby fittingly illustrating the absence of an effect of treatment on mean values detected by our analyses printed in Table 1.

In Figure 2b, we now fittingly illustrate the main finding of our study, showing visually that fast explorers breed relatively late versus early when perceived predation levels are low versus high. Visually illustrating these patterns is challenging as plots of the “raw” data (requested by the referee) are uninformative in this particular case. Specifically, plots of raw data depict patterns of so-called “raw” phenotypic correlations that are known to give biased (in this case attenuated, see Methods) estimates of the among-individual correlations of interest here (see Eqn. 6 in Dingemanse & Dochtermann 2013, *J Anim Ecol* 82: 39-54). Among-individual correlations used as input for our path models are sometimes visualized by plotting correlations between individual-mean values, but unfortunately correlations between individual-mean values are also biased (towards residual within-individual correlations; see equations in the Appendix of Dingemanse et al. 2012, *Behav Ecol SocioBiol* 66: 1543-1548). We therefore fittingly chose to provide plots of each individual’s Best Linear Unbiased Predictors (BLUPs) for lay date (y-axis) and exploration score (x-axis). These BLUPs represent our best estimate of an individual’s average value for the two focal traits corrected for the sample size per individual. Reassuringly, these plots neatly match our statistical analyses based on which we conclude that the relationship between lay date and exploration score is opposite between low and high predation level treatment groups.

Second, the authors rely on clutch size as a proxy for fitness – this needs to be justified. Is a larger clutch size always great fitness, and might that depend on perceived predation risk? What if females prioritize quality of eggs over quantity of eggs?

Response: The referee asks a very important question here. Fortunately, in this case, literature on passerines and other birds shows that clutch size is a reliable indicator of reproductive fitness (e.g. Tinbergen and Daan 1990, *Behaviour* 114:1-4; Pettifor et al 2008 *J Anim Ecol* 70:62-79). The key question here is whether the lack of difference in clutch size between behavioral types that we detected in both treatment groups really implies that the behavioral types had equal reproductive success. To address this question forcefully, we therefore re-ran our analyses using the number of fledglings (instead of clutch size), a key down-stream integrative fitness trait, as a response variable. This analysis demonstrated that the number of fledglings did not differ between behavioral types in either treatment group (low-PPL: 0.04 (-0.06,0.16), $p = 0.43$; high-PPL: 0.07 (-0.04, 0.19), $p = 0.23$). This key finding implies that the types indeed did not differ in reproductive fitness in either treatment group. See text additions lines 209-211.

Here are more minor comments:

Line 28: plasticity of what?

Response: We have added 'in reproductive behavior'

Line 49: "starting to breed"

Response: We have changed as reviewer suggested

Line 51: choose another word besides "down-regulating" – something like decreasing (same for "up-regulating")

Response: We have changed to decreasing and increasing, as per reviewer's suggestion

Line 60: omit "of"

Response: Done

Line 64: I know it's in the methods but tell the reader here how often and for how long you broadcast the songs. Although it looks as though the authors took measures to prevent habituation of the great tits to the songs of the sparrow hawks, did the treatment regime reflect the biology of sparrow hawks? For example calls were played for 4 consecutive days then not played for the next 4 consecutive days. Is this how sparrowhawks would normally behave? What information is this experimental regime providing to the great tits about danger? Is the treatment manipulating risk or is it more influencing uncertainty about risk? The authors clearly assume the former – that they are manipulating perceived risk – but it would be worthwhile to consider the alternative, i.e. that the "high risk" treatment was really the "uncertain" treatment.

Response: We have added a short description of the playback scheme here as requested (line 67 in revised MS). Additionally, when sounds were broadcast the frequency of vocalizations matched natural sparrowhawk vocalization behavior. The 4 day on 4 day off scheme does of course not exactly match sparrowhawk natural behavior- they are not quite so regular in their movements. But, sparrowhawks do hunt over relatively large territories, and they are silent when actively hunting, so it is not unusual for there to be extended periods (up to days) where there are no vocalizations, but yet the risk of sparrowhawk predation persists. Therefore, not hearing a sparrowhawk for a few days should not alter a great tits perception of overall risk in the area. Previous work in our study area confirms this, as birds sang less and had a higher ratio of alarm calls to songs, even on days our speakers were not broadcasting (Abbey-Lee et al. 2016, Behav Ecol 27:708-716). Additionally, previous research in other species implies that a single observation of a predator has a long-lasting effect. Animals don't seem to really use the variance, but rather assume presence for a long time if there is any signal of a predator (see e.g. Gabriel et al. 2005, Am Nat 166: 339-353). See in text additions lines 287-295.

Line 75: “game”? Do you mean “strategy”?

Response: We have changed to match reviewer’s suggestion

Line 115: remove “,” after “Either”

Response: Done

Reviewer #2 (Remarks to the Author):

This novel and impressive study employed a large-scale manipulation to test how environmental variation drives plasticity in reproductive strategies, and whether plasticity differs by personality type. The authors found that in great tits, perceived risk of predation influenced variation in breeding phenology, and that this variation was explained by personality type (exploratory speed). The experimental design is elegant, and the sample sizes were sufficiently large to enable estimation of within and among individual variation. The results provide important new insights into the evolution of personality, and how the risk of predation generates variation in plasticity.

Response: We are grateful to learn that the reviewer finds our study novel and impressive. We have fully addressed the reviewer comments as detailed below.

The one major suggestion that I have is to more clearly differentiate between the various aspects of reproductive investment and breeding decisions that were measured, and their potential fitness outcomes. In the discussion, clutch size, which is framed as a measure of reproductive investment (e.g., line 114) is conflated with reproductive success (also defined as clutch size on line 192). Conceptually, this is problematic because the premise of this study rests on the idea that the offspring of late breeders will face a greater risk of predation. Because of this, clutch size will not be a good measure of reproductive success, since the fitness benefit of, for example, a 5-egg clutch laid early in the season will on average be higher than that of a 5-egg clutch laid later in the season, at least in the presence of predators.

Because this experiment separated perceived risk of predation from actual predation risk, the actual fitness cost of breeding in a high-predation environment will not have been realized by individuals in the high-PPL group. Thus, their reproductive success may track more closely with their clutch size than individuals in a truly high-predation environment.

We fully agree with the referee that actual and perceived predation risk can have very different fitness consequences. We have rephrased parts of our main texts to clarify that our actual interest here is in addressing how behavioral types differ in how they adjust life-history decisions to perceived risk strategically (lines 70-72). Further studies are indeed required to address the fitness consequences of these strategic adjustments, an issue that we now bring forward to our General Discussion (lines 231-236).

But regardless, distinguishing between reproductive investment and reproductive success in interpreting these results is important for understanding how fluctuating selection on plastic

responses to environmental conditions may maintain distinct personalities. It seems that fledging success was measured in this study (lines 227-228); it would also be nice to see those relationships discussed here.

Response: We fully addressed this important issue, which was mentioned by Reviewer 1. See our detailed response above. Briefly, we agree that reproductive investment (clutch size) does not necessarily equate reproductive success (fledging number). However, as detailed above (see our responses to Reviewer 1), our conclusions stand when we swapped clutch size for fledging number. See text additions line 193-195.

Additionally, while I realize the manuscript has tight length constraints, I think a bit more discussion in two areas would help to place these findings in context more clearly. First, faster-exploring birds are described as having a faster pace-of-life, based on previous work. But the results shown here don't seem to be entirely consistent with this. Faster explorers lay more eggs than slow explorers breeding on the same date, but if I'm interpreting these results correctly, they don't lay more eggs overall – suggesting that their annual reproductive investment is not higher on average. Perhaps they do invest more at later stages of breeding (and achieve higher average annual reproductive success, offsetting their shorter lifespans), but this isn't clear from the discussion presented here.

We agree with the reviewer. The above-mentioned correlations between exploration score and fledging number are not significant in either treatment group, thus exploration types don't differ in pace of life (i.e. in our measure of annual reproductive investment) in our study. Additionally, the cited evidence for POLS used in the previous version of the MS could also be explained by other mechanisms, therefore we have altered our discussion about POLS to conclude more forcefully that the current study does not actually confirm pace-of-life predictions as faster explorers don't produce more eggs overall. See lines 222-224.

Second, I think a bit more discussion on why slow explorers might delay their average lay date (as opposed to just their lay date relative to fast explorers) under high-PPL would improve the clarity. Competition is briefly mentioned in general terms on line 214, but is there evidence that resource competition between slow and fast explorers is a likely driver of the delay in lay date amongst slow explorers in high-PPL?

Response: We have now expanded our discussion about resource competition between slow and fast exploring great tits in our Discussion, lines 216-218. We know that faster explorers are more dominant at clumped food resources (Dingemanse & de Goede 2004, Behav Ecol 15:2023-1030). Therefore it may be more important for slow explorers, the weaker competitors, to shift their breeding (and thus time of peak fledgling feeding requirements), particularly in high-PPL scenarios when the benefits of breeding early are diminished.

A few other minor suggestions:

- The scenarios in Figure 1c all keep the mean breeding date similar under both contexts, and the variance either the same or higher under high-PPL. Of course it isn't possible to graphically illustrate all possible scenarios, but I would recommend stating that this figure illustrates possible

scenarios given those assumptions. The analyses done should enable the authors to distinguish among these and other scenarios (e.g., reduced variance in lay date under high PPL or an overall advancement in lay date), so this is not at all a criticism of the analytical framework, simply a suggestion to clarify the possible patterns.

Response: We have added a line to the figure legend to address this point.

- The term “April date” is used on line 98 – something like “days since April 1” would be clearer

Response: We have updated to match the reviewer suggested phrasing.

- The rates at which predator vocalizations were broadcast seems high – do sparrowhawks really vocalize that much (lines 253-256)?

Response: Yes, our vocalization scheme was based on data from birder/sparrowhawk experts in northern Europe.

- Line 293: It looks like “was” should be deleted from the sentence “...between the full model and a model with the focal random effect removed”

Response: Agreed, we have updated based on reviewer’s suggestion.

- Line 392: “is” should be “are”

Response: Done

- Line 193: “did not vary as a function”

Response: Done

REVIEWERS' COMMENTS:

Reviewer #1 (Remarks to the Author):

The authors did a good job responding to the concerns that were raised during the last round of review. They were pretty quick to dismiss my point about whether their experimental manipulation of perceived predation risk influenced certainty, relying on a model by Gabriel et al 2005 which found a stronger influence of means than variances in risk. However, there is a large literature on the risk allocation hypothesis (sensu Lima and Bednekoff 1998) supporting the idea that temporal variance in risk is actually key to understanding anti predator behaviour. I don't think this is a lethal point that is going to sink the entire paper and I appreciate that the paper now includes more discussion of their particular manipulation.

It also appears that there are still some places in the MS that use the term down-"regulate", e.g. lines 58,69.

Reviewer #2 (Remarks to the Author):

The authors' response and revisions were very thorough, and sufficiently addressed all of my previous comments. I have no further suggestions to make.

REVIEWERS' COMMENTS:

Reviewer #1 (Remarks to the Author):

The authors did a good job responding to the concerns that were raised during the last round of review. They were pretty quick to dismiss my point about whether their experimental manipulation of perceived predation risk influenced certainty, relying on a model by Gabriel et al 2005 which found a stronger influence of means than variances in risk. However, there is a large literature on the risk allocation hypothesis (sensu Lima and Bednekoff 1998) supporting the idea that temporal variance in risk is actually key to understanding anti predator behaviour. I don't think this is a lethal point that is going to sink the entire paper and I appreciate that the paper now includes more discussion of their particular manipulation.

We are happy to hear that we have reviewer 1 appreciates our modifications. We have now added further discussion of the risk allocation hypothesis and how temporal variance is related to anti predator behavior, see lines 367-370.

It also appears that there are still some places in the MS that use the term down-"regulate", e.g. lines 58,69.

We have replaced down-regulated with decreased in these two instances, we found no further use of the terms down or up regulated in the MS.

Reviewer #2 (Remarks to the Author):

The authors' response and revisions were very thorough, and sufficiently addressed all of my previous comments. I have no further suggestions to make.

We are happy to hear we have satisfied reviewer 2.